# Plasma microRNAs as a Potential Biomarker for Identification of Progressive Supranuclear Palsy

**DOI:** 10.3390/diagnostics12051204

**Published:** 2022-05-11

**Authors:** Palaniswamy Ramaswamy, Rita Christopher, Pramod Kumar Pal, Monojit Debnath, Ravi Yadav

**Affiliations:** 1Department of Neurology, National Institute of Mental Health and Neuro Sciences (NIMHANS), Bengaluru 560029, India; palani.swamy20@gmail.com (P.R.); palpramod@hotmail.com (P.K.P.); 2Department of Neurochemistry, National Institute of Mental Health and Neuro Sciences (NIMHANS), Bengaluru 560029, India; rita.nimhans@yahoo.com; 3Department of Human Genetics, National Institute of Mental Health and Neuro Sciences (NIMHANS), Bengaluru 560029, India; monozeet@gmail.com

**Keywords:** biomarker, diagnosis, microRNA, plasma, supranuclear palsy

## Abstract

Progressive supranuclear palsy (PSP) is the second most common Parkinsonian disorder with complex etiology. The underlying molecular mechanism of PSP pathogenesis remains unclear. The present study aims to find the feasibility of using plasma miRNAs as novel biomarkers. Plasma-focused qPCR panels were used for microRNA profiling and identified differentially expressed microRNAs in PSP compared to controls. The DIANA-miRPath v3.0 was used to perform KEGG pathway analysis. We then confirmed the expression of selected candidates by RT-qPCR and their clinical utility was assessed by ROC analysis. Profiling data revealed 28 differentially expressed microRNAs in PSP. Five overexpressed miRNAs were selected for further analysis. The KEGG pathway analysis revealed 48 high-risk pathways. The study revealed that as a single marker—miR-19b-3p, miR-33a-5p, miR-130b-3p, miR-136-3p, and miR-210-3p had a specificity of 64.71%, 82.35%, 68.75%, 82.35%, and 70.59% at sensitivity 77.78%, 77.78%, 66.67%, 73.33%, and 66.67%, respectively. The result suggests that circulating plasma miRNAs were altered in PSP compared to control. The findings of this study may provide potential biomarkers and pathways associated with PSP. Further large-scale validation studies are required to confirm the same.

## 1. Introduction

Progressive supranuclear palsy (PSP) is one of the most frequent atypical Parkinsonisms that affects 5–10/100,000 people, with no apparent variation across geographical locations, ethnicity, or race (aged > 60 years) [1]. The core characteristic clinical symptoms of PSP are supranuclear gaze palsy, dysarthria, dysphagia, and frontal cognitive syndrome [2]. The major pathological hallmarks are globose neurofibrillary tangles due to excessive accumulation of hyperphosphorylated tau protein in neurons and glia across several brain regions such as the cerebral neocortex, subthalamic nucleus, periaqueductal grey matter, substantia nigra, pallidum, superior colliculi, and dentate nucleus [3]. Though the definitive diagnosis of PSP can be established neuropathologically by post-mortem, in the majority of the cases, the diagnosis is made based on the movement disorder society criteria [2]. The precise diagnosis of PSP in the early phase of the disease is difficult due to the presence of overlapping symptoms with several other conditions [4,5]. Furthermore, there are several subtypes of PSP (PSP-RS, PSP-P, PSP-PGF, PSP-SL, PSP-F, and PSP-C), which makes the diagnosis even more challenging [4].

Over the past few years, there have been several attempts to understand the genetic basis of PSP and also to identify biomarkers that could aid in establishing the diagnosis and assessing the progression of PSP. Few studies have suggested the level of serum/plasma neurofilament light chain or tau forms as markers of diagnosis and disease severity [6,7,8]. Further, Positron Emission Tomography (PET) imaging of 18F-PI-2620 distribution volume ratios has been suggested as biomarkers for PSP [9]. In some studies, genetic risk determinants, cognitive profile, clinical phenotypes, brain imaging correlates, serum level of neurofilament light chain, etc. have been used in distinguishing the clinical subtypes of PSP and other atypical parkinsonian syndromes [4]. The serum neurofilament light chain and tau protein quantification lack sensitivity and specificity. The tau level in cerebrospinal fluid (CSF) is too low and it has limited application due to the invasiveness of the test. Although PET imaging offers high sensitivity, its use is restricted due to its invasive nature and high cost. There exists a lack of reliable blood-based biomarker(s) that can potentially facilitate the early detection of PSP and its subtypes.

A growing body of evidence suggests that microRNAs (miRNAs or miRs), a class of non-coding endogenous single-stranded, small (18–25 nucleotide) RNAs that regulate gene expression at the post-transcriptional level in many neurodegenerative diseases [10], have important implications. The miRNAs bind to the complementary sequence of messenger RNAs at the 3’-untranslated region and regulate gene expression through translation inhibition or degradation. MiRNAs play pivotal roles in neurobiological processes that include cell differentiation, proliferation, neuronal patterning, and cell survival [11]. Notably, dysregulation of miRNAs has been linked to many neurodegenerative diseases, and they are emerging as reliable biomarkers of several neurodegenerative diseases, such as Alzheimer’s disease [12], Parkinson’s disease, and multiple system atrophy [13]. MiRNAs are increasingly being proposed to play a key role in tauopathies [14]. However, studies on miRNAs are albeit limited for PSP. In a previous study on the post-mortem brain, highly increased expressions of miR-147a and miR-518e were reported [15]. In another study, miR-132 was found to be down-regulated in the brains of PSP patients and this was associated with tau exon 10 inclusions [16]. The blood profile of miRNAs and their relevance to the risk and progression of PSP are inadequately known. To address this knowledge gap, the current study was aimed at identifying circulatory plasma miRNAs in PSP.

In this study, we examined differentially expressed plasma miRNAs of PSP compared to healthy participants. Subsequently, the functions, pathways, and common hub genes of candidate miRNAs were predicted by bioinformatics analysis. The feasibility of using plasma miRNAs as novel biomarkers for PSP was evaluated by quantitative real-time PCR and ROC analysis.

## 2. Materials and Methods

### 2.1. Study Participants

In this prospective study, we randomly recruited 18 patients (aged 60.11 ± 1.6 years) with probable PSP within three years of the clinical diagnosis from the Outpatient Services of the Department of Neurology, National Institute of Mental Health and Neurosciences (NIMHANS), Bangalore, from July 2017 to March 2020. Movement Disorder Society diagnostic criteria [2] were used to establish the diagnosis of patients with PSP. The cognitive function in these patients was assessed by the Mini-Mental State Examination (MMSE). Only patients with an MMSE score ≥ 24 for literates and ≥21 for illiterates suggestive of cognitively unimpaired were included in the study. Normal healthy individuals (n = 17) matched for age (57.8 ± 1.4 years) were recruited. From this population, we selected 12 PSP patients and 9 controls as the initial screening cohort (Appendix A). Ethical approval was obtained from the Institutional Ethics Committee of NIMHANS and written informed consent was obtained from all the participants.

### 2.2. Collection of Blood Samples and Processing

Fasting venous blood (5 mL) was collected by venepuncture from all study participants and transferred to EDTA vacutainers for further processing. The EDTA treated blood sample was centrifuged at 3000 rpm for 15 min at room temperature to separate blood cells from the liquid part of the blood called plasma. The separated plasma was collected carefully without disturbing the buffy coat and, again, the plasma sample was centrifuged to minimize buffy coat contamination and then aliquoted for further study. The aliquoted plasma sample was stored at −80 °C or used immediately for miRNA isolation. The entire process of collecting, separating, and storing plasma samples was completed within 45 min.

### 2.3. Plasma miRNA Extraction and cDNA Synthesis

MicroRNAs were extracted using the miRNeasy Serum/Plasma Advanced Kit (50) with spike-ins (Catalog # 217204, Qiagen, GmbH, Hilden, Germany) as per the manufacturer’s instructions. While extracting the sample, 1 µg of carrier RNA was added (MS2 RNA, Roche, Basel, Switzerland) to each sample. The extracted miRNAs were then reverse transcribed using the miRCURY LNA RT Kit (Catalog # 339340, Qiagen, GmbH, Hilden, Germany) in a total of 10 µL volume reactions containing 4 µL of isolated miRNAs as per the manufacturer’s instruction. For qPCR amplification, 3 µL of 20× diluted cDNA was assayed in a 10 µL PCR reaction using miRCURY LNA SYBR Green PCR master mix (Catalog # 339345, Qiagen, GmbH, Hilden, Germany). The plasma samples of the validation cohort were processed similarly.

### 2.4. Plasma miRNAs Profiling by qPCR

For the miRNA screening, miRCURY LNA miRNA Focus PCR Panels (Catalog # YAHS-106Y-8, Qiagen, USA) were used. Each panel consists of pre-coated primers for 179 miRNAs, including cel-miR-39-3p, UniSp2, UniSp4, UniSp5, and UniSp6. In addition, each 96-well PCR panel contained an inter-plate calibrator in triplicate and an empty negative control. The miRNA profiling experiment consisted of pooled samples from PSP patients, as well as healthy subjects in triplicate (Appendix A). The qPCR reaction was performed as described above using the ABI 7500 Real-Time PCR System (Applied Biosystem, Waltham, MA, USA). The SDS v2.3 programme was used to evaluate the amplification, Ct-value, and melting curve analysis curves (Applied Biosystems, Waltham, MA, USA). According to instructions of QIAGEN Inc., the Ct-values of all miRNAs were submitted to the GeneGlobe web tool (https://dataanalysis2.qiagen.com/ accessed on 11 December 2019) and followed onscreen instructions to identify differentially expressed miRNAs.

### 2.5. Functional Analysis and Risk Module Identification

A number of computational methods for identifying miRNA target genes have already been developed. In our study, the target genes of all five candidate miRNAs were predicted using miRDB (http://mirdb.org accessed on 22 August 2021), TargetScan v7.2 (http://www.targetscan.org/vert_72/ accessed on 22 August 2021), and miRWalk (http://mirwalk.umm.uni-heidelberg.de/ accessed on 22 August 2021) individually. Only genes recorded for each of the miRNAs in three databases were considered as ultimate target genes of that particular candidate miRNA, and they were mapped to the miRPath v3.0 tool (http://snf-515788.vm.okeanos.grnet.gr/ accessed on 23 August 2021) to identify risk pathways and associated genes [17]. The *p*-value of <0.05 was set as a threshold. Risk pathways regulated by at least two miRNAs were identified and enlisted.

### 2.6. Confirmation of Candidate miRNAs Using qPCR

Candidate miRNA expression (miR-19b-3p, miR-33a-5p, miR-130b-3p, miR-136-3p, and miR-210-3p) was measured in triplicate using qPCR, with miR-16-5p serving as an endogenous reference. Appendix A lists the primer’s specifics. As a reference sample, a pooled control sample was used. Amplification was carried out as stated above, and data were expressed as fold change.

### 2.7. Statistical Analysis

GraphPad Prism v.6.0 was used for statistical analysis (GraphPad Software, Inc., San Diego, CA, USA). The nonparametric Mann–Whitney test was used to compare the data of two groups, with *p* < 0.05 being regarded as statistically significant. The Holm–Sidak technique was used to compensate for multiple comparisons. The clinical usefulness of candidate miRNAs was assessed using receiver operating characteristic (ROC) curves. Data are given as means with standard error (S.E).

## 3. Results

### 3.1. Demographic Variables

Appendix A provides the clinical and demographic data of the study participants in the initial screening and validation cohorts, respectively. There were no significant differences in sex ratio or age between PSP patients and control groups.

### 3.2. Differential Expression of miRNAs in Plasma and Identifying Candidate miRNAs

MiRNA profiling demonstrated a good signal-to-background ratio for 120 miRNAs in both samples. The principal component analysis (PCA) was performed to know the similarity between the miRNA expression patterns in PSP and control groups (Figure 1a), and the result demonstrated that they are different. The variance in plasma miRNA expression between the groups was assessed using a scatter plot (Figure 1b) and a volcano map (Figure 1c). Up-regulated miRNAs are depicted in red and down-regulated miRNAs are presented in green when compared to controls (Figure 1b,c). A total of 28 miRNAs were identified as being differentially expressed in PSP patients compared to controls (*p* < 0.05 with a fold change of <1.5 or >1.5) were enlisted (Appendix A). The hierarchical clustering analysis of 28 differentially expressed miRNAs was displayed using a heat map, which showed that 23 miRNAs were upregulated and five were downregulated compared to control, respectively. Low to high expression of each miRNA transcript was represented by the color scale, which ranged from green to red (Figure 2). We chose five upregulated miRNAs for validation (i.e., miR-19b-3p, miR-33a-5p, miR-130b-3p, miR-136-3p, and miR-210-3p). Appendix A lists the primers for all of these miRNAs.

### 3.3. Validation of Candidate miRNAs Using RT-qPCR

Synthetic spike-ins were utilized to test the quality of plasma miRNA isolation (UniSp2) and cDNA synthesis efficiency (UniSp6). The average threshold cycle (Ct) values of the spike-ins in all the samples were 20.02 and 18.92, respectively, with a low standard deviation of 0.02 and 0.06, indicating that the amount of miRNA in each sample examined in the qPCR was identical. Two miRNAs, miR-451a and miR-23a, were used to measure hemolysis in all samples. The Ct difference between these two miRNAs was <5, suggesting that the samples had not been hemolyzed and were therefore appropriate for further investigation.

The expressions of selected candidate miRNAs were tested in 17 healthy controls and 18 PSP patients who were enrolled in the study individually. The findings indicated that putative candidate miRNAs were overexpressed in plasma samples of patients, and they complemented the initial screening data (Table 1). The ROC curve analysis was used to assess the potential of a candidate miRNA as a biomarker for distinguishing PSP patients from healthy controls. With AUCs of 0.7059, 0.8578, 0.7778, 0.7882, and 0.7810, the candidate miRNAs miR-19b-3p, miR-33a-5p, miR-130b-3p, miR-136-3p, and miR-210-3p displayed excellent specificity and sensitivity (Table 1). The performance of all five potential miRNAs in multi-marker diagnostics was then investigated. The combined ROC of these five miRNAs has an AUC of 0.7817 with a sensitivity of 72.41% and a specificity of 66.67% with a 95% CI (0.7126 to 0.8508).

### 3.4. Kyoto Encyclopedia of Genes and Genomes (KEGG) Pathway Enrichment Analysis and Identification of Hub Genes

Every miRNA can interact with multiple genes. The target genes of candidate miRNAs selected for validation were predicted and verified by miRDB, TargetScan, and miRWalk. The common target genes regulated by each candidate miRNA in all the databases were used for identifying pathological pathways associated with PSP. Then, we performed the KEGG pathway analysis for the candidate miRNAs to learn about the miRNA-mRNA network using the online tool—DIANA miRPath v3.0. This tool detected 48 KEGG pathways with *p* < 0.05, which are listed in Appendix A. We limited risk pathways even more by finding target genes present in multiple pathways that were affected by two or more candidate miRNAs, which we dubbed “hub genes.” As a consequence, 12 different risk pathways and associated hub genes have been determined (Appendix A). The following are the genes that have been identified as common hub genes: IRS2, CCNB1, PIK3CB, MAPK14, CCND2, RAF1, CDKN1B, SMAD4, SMAD5, MAPK8, PRKAA1, CDKN1A, PTEN, TGFBR2, BCL2L11, SOS1, GSK3B, MYC, CCNA2, CHEK1, FZD5, INHBA, ACVR1, PPP2R5E, HSP90B1, CANX, DNAJA1, SAR1B, HERPUD1, VCP, UBE2D3, ZMAT3, CASP3, BAMBI, CALM2, SREBF1, FRMD6, PARD6B, ZFYVE9, SP1, UBQLN2, SEC24C, and PPP1CB. The hub genes may be involved in PSP etiology or progression.

## 4. Discussion

The most prevalent form of atypical Parkinsonism is PSP, which is a rare neurological illness. Because the symptoms of PSP are similar to those of other movement disorders [1,5], it is difficult to diagnose. Notably, the onset and intensity of cardinal symptoms differ significantly between PSP patients and their subtypes. Current clinical diagnostic methods lack diagnostic accuracy and reliability at the early stages of the disease due to overlap of clinical symptoms of PSP and similar pathologies and mild initial responses of PSP patients to dopaminergic treatment [18]. The protein biomarkers identified earlier differentiate between PD and other Parkinsonisms, but cannot selectively recognize PSP [19,20,21]. For the exact diagnosis and subtyping of PSP, a more reliable, non-invasive, cost-effective, and disease-relevant biomarker is required. MiRNAs are relatively stable in bodily fluids and may be easily tested using non-invasive techniques. [22]. An emerging body of recent studies suggests that miRNAs have the potential to serve as biomarkers of Parkinson’s and related disorders (reviewed in [13]). The number of research studies on the miRNA profile in PSP and its subtypes remain limited [15,16]. However, the role of circulating miRNAs in patients with PSP is inadequately known.

The current study discovered changes in the expression of 28 miRNAs in PSP, with 23 being upregulated and 5 being downregulated. Five of the 23 increased miRNAs were chosen for further study: miR-210-3p, miR-19b-3p, miR-33a-5p, miR-130b-3p, and miR-136-3p. There have been few results of miRNA profiles in PSP, and these miRNAs have not been documented in prior PSP investigations. Previous studies on post-mortem brain tissues discovered that miR-132 was downregulated whereas miR-147a and miR-518e were upregulated [15,16]. A microRNA analysis in the CSF of PSP patients by Nanoka et al. revealed downregulation of miR-873-3p and miR-6840-5p and upregulation of miR-204-3p [23]. Whereas Starhof et al. have demonstrated that miR-106b-5p, miR-145-5p, and miR-204-5p were downregulated, and miR-184 and miR-30b-5p were upregulated in CSF of PSP patients compared to control [24], as well as miR-106b-5p provided the best discrimination between PD and PSP [24]. Manna et al. proposed three exosomal miRNAs, miR-425-5p, miR-21-3p, and miR-199a-5p that distinguished PSP from PD [25]. However, most of the studies discussed above identified very different sets of differentially expressed miRNAs in PSP without any overlap. This could be due to a difference in the methodology they used. The age of the study groups and the types of the samples might have also played a major role in the non-overlap of the candidate miRNAs. The reasons for the minimal overlap between the studies have been discussed and reviewed earlier [13,26] Therefore, there is a need to have a consensus on the standard operating procedures used for identifying miRNA-based biomarkers.

In this study, we describe a network of miRNAs—target genes—signaling pathways in PSP, which may aid in understanding the etiology of PSP and associated diseases. Our study revealed that the candidate miRNAs might be linked to a number of hub genes. Notably, when we compared the target genes of miR-132, miR-147a, and miR-518e with the Appendix A, miR-132 targeted SMAD5, CDKN1A, PTEN, BCL2L11, SOS1, GSK3B, ACVR1, PPP2R5E, and SREBF1, while miR-147a targeted MAPK14, SMAD4, SMAD5, MAPK8, PRKAA1, CDKN1A, PTEN, TGFBR2, SOS1, MYC, INHBA, UBE2D3, ZMAT3, BAMBI, and UBQLN2, whereas miR-518e regulated only the ZMAT3 gene. Assuming the association, these discovered hub genes ought to be involved in high-risk pathways that could contribute to the etiology and development of PSP. Our research uncovered a total of twelve high-risk routes. The functions of these hub gene-controlled pathways include neural progenitor cell maintenance, differentiation, and oxidative stress resistance, as well as neuronal cell survival, longevity, inflammation, autophagy, and apoptosis.

Transforming growth factor (TGF)-β pathway and FoxO signaling pathways have been implicated in many cellular processes and modulated by at least three of the identified candidate miRNAs (Appendix A). Thus, these pathways might be considered high-risk signaling for PSP development and/or progression. In silico miR-19b-3p, -33a-5p, -130b-3p, and -136-3p were shown to have the ability to influence the TGF pathway by affecting four conserved upstream genes and five downstream genes. Extracellular neurotrophic factors and associated signals are required for the survival of neurons and their progenitors [27]. The TGF pathway synergizes with neurotrophins, and diminished TGF signaling has been linked to neuronal death and neurodegeneration [20]. Furthermore, in PSP, phosphorylated Smad2/3 colocalizes with phospho-tau inclusions [28]. This points to TGF-β’s significance in PSP pathobiology. In mature neurons, FOXO regulates age-dependent axonal degeneration [29] and in neuronal stem cells, it is essential for the maintenance of stem cell quiescence and ROS clearance [21]. Additionally, FOXOs are well-known MST1 substrates in neurons. When primary mammalian neurons are exposed to oxidative stress, MST1 phosphorylates FOXO proteins and increases FOXO nuclear translocation, resulting in neuronal death [30]. This demonstrates that FOXO also regulates Hippo signaling and is important in the regulation of cell proliferation, differentiation, and oxidative stress-induced neuronal cell death [30,31]. However, the exact role and regulatory mechanism of these signaling in PSP remains unclear. They may be related to PSP in terms of neuronal cell survival and anti-stress properties. However, a greater understanding of these signaling target specificity and their relation to PSP is needed.

The upregulation of miR-210-3p was the most notable discovery of the current investigation. It is worth noting that miR-210-3p can target several genes involved in cell division, migration, mitochondrial metabolism, angiogenesis, DNA repair, and chromatin remodeling, among other cellular processes [32]. MiR-210 has been identified as a hypoxia-regulated, highly conserved miRNA. Its role in cancer has been extensively studied, and it has been discovered to behave as a tumor suppressor or onco-miR. The miR-210-3p was shown to activate the NF-kB signaling pathway by targeting numerous negative regulators of the NF-kB signaling pathway [33]. This indicates that miR-210-3p plays a critical role in immune-inflammatory responses. MiR-210 has also been shown to target numerous genes involved in neuronal plasticity [34], and ectopic overexpression in the mouse brain increases neural progenitor proliferation [32]. In recent research, hypoxia-induced miR-210 was shown to target synaptic function genes (GRINA, TMUB1, and AP2S1), plasticity genes (ACTB), Alzheimer’s disease genes (APOE), and genes implicated in MAPK/VEGF and OXPHOS signaling pathways [34]. This shows that miR-210 has a crucial significance in the neurodegenerative process. According to these findings, increased levels of miR-210-3p might play a role in the etiology of PSP by changing immune-inflammatory pathways and neural plasticity.

Another intriguing discovery was increased circulating levels of miR-19b-3p in PSP patients. Changes in this miRNA’s expression have previously been linked to a variety of neurodegenerative disorders, including tauopathies. The serum levels of miR-19b-3p were shown to be downregulated in PD patients with LRRK2A G2019S mutations [35], whereas plasma levels were raised [36]. MiR-19b-3p was also shown to be downregulated in the CSF and plasma of individuals with multiple system atrophy [37], and lower serum levels of miR-19b-3p have been suggested as a biomarker for Alzheimer’s disease [38]. Several genes involved in cell proliferation, differentiation, and immunological responses are targeted by miR-19b-3p. In silico, the 3’-UTR of STAT3 was identified as the target of miR-19b-3p. STAT3 activation causes Th17-mediated immune-inflammatory responses [39]. By targeting ring finger protein 11, a negative regulator of NF-kB signaling, miR-19b-3p was shown to regulate Japanese encephalitis virus-induced inflammation [40]. When miR-19b-3p was overexpressed, it increased the production of IL-6, IL-1, TNF-α, and chemokine ligand 5 [40]. Furthermore, miR-19b-3p has been demonstrated to modulate *E. coli*-induced neuroinflammation by targeting TNFAIP3, a negative regulator of NF-kB activation [41]. A recent study found increased expression of miR-19b-3p in the prefrontal cortex of individuals with severe depressive illness, and it was postulated that this modulates target genes involved in synaptic plasticity [42]. These data imply that this miRNA is involved in both neuronal and immune-inflammatory processes. Given the importance of microglia activation and consequent neuroinflammation in PSP, miR-19b-3p appears to play a major role in modulating neuroinflammation.

In PSP patients, circulation levels of miR-33a-5p were also found to be considerably greater. The miR-33 family of miRNAs, which includes miR33a and miR-33b, has several target genes and is primarily engaged in modulating the expression of genes that regulate cholesterol and fatty acid production [43]. Notably, they regulate cholesterol metabolism in glial and neuronal cells by targeting the ATP-binding cassette transporter A1 (ABCA1) gene [44]. Furthermore, miR-33a-5p regulates p53 and downregulates neural genes during forebrain development by modifying FoxG1 [45,46]. MiR-33 loss affects the location of peripheral inflammatory monocytes [47] and reduces inflammation by modifying numerous anti-inflammatory pathways [48]. Based on the information presented above, miR-33a-5p might have a role in PSP pathobiology by affecting brain cholesterol metabolism and immunological inflammation.

PSP patients had considerably greater circulating levels of miR-130b-3p than controls. Multiple genes are targeted by miR-130b-3p. Certain tumors were found to have high levels of miR-130b-3p. Upregulation of miR-130b promotes the viability of prostate cancer cells while inhibiting apoptosis [49]. PTEN has been identified as a target of miR-130b-3p in cancers, where it appears to enhance proliferation, migration, and invasion by downregulating PTEN via PI-3K expression [50]. On the other hand, miR-130b-3p is downregulated in Alzheimer’s patients [51]. MiR-130b-3p has recently been implicated in the control of immunological responses, and it has been shown to influence M1 macrophage polarization by targeting Interferon Regulatory Factor-1 [52]. Furthermore, altered expression of miR-130b was seen in oxidatively challenged primary hippocampal neurons, suggesting a function for this gene in neurodegenerative disorders [53]. The foregoing findings imply that miR-130b-3p has a crucial significance in immunological homeostasis and brain functioning and that it can cause PSP by triggering immune-inflammatory responses.

In PSP patients, circulating levels of miR-136-3p were also increased. PTEN and Kruppel-like factor 7 (KLF7) are two of miR-136-3p’s target genes [54,55]. KLF7 is an oncogene that is adversely controlled by miR-136-3p in glioma and other malignancies [46]. It also regulates the immune system by regulating IL-17. Overexpression of miR-136-5p led to increased production of inflammatory factors and modulated inflammatory pathways during spinal cord injury [56]. Based on these findings, miR-136-5p may play an important role in PSP by modulating immune-inflammatory responses. According to the available research and in silico projections, identified candidate miRNAs are mostly involved in the control of immune-inflammatory responses as well as neurological functions. The ROC analysis in this study revealed that these miRNAs might be considered promising biomarkers for PSP.

The miRNA databases employed for bioinformatics analysis are the study’s principal shortcoming. Because these databases may have restricted data sources, our study may need to be modified in the future as databases become more comprehensive and revised. Second, the present sample size of 35 participants is insufficient for a conclusion to be drawn. We chose sample pooling for initial miRNA screening because it decreased costs, shortened analytical run times, and compensated for the limited number of plasma samples available, allowing us to profile a significant number of miRNAs in both patient and control samples at a low cost. Pooling of biological specimens has been used in earlier studies to identify miRNA as a biomarker [57]. Although sample pooling has its advantages, it is not the best technique since it does not account for biological heterogeneity within individual samples while reducing information loss. The current study looked at plasma miRNA rather than CSF miRNA. CSF may be a better way to look at neurodegenerative diseases like PSP. However, because getting plasma samples is easier than collecting CSF, investigations of plasma miRNA expression in individuals with PSP may be more appropriate.

## 5. Conclusions

The most frequently used diagnostic approaches for PSP are neuroimaging and rating scales. PSP diagnosis based on these methods is challenging and limited due to their lack of usefulness in a wide clinical environment. To address this, we employed miRNA profiling and identified five plasma miRNAs that are overexpressed in PSP patients compared to healthy controls. At present, this miRNA panel has the potential to be used as a secondary diagnostic tool for detecting PSP in its early stages with more confidence. The identified miRNAs regulate a variety of important cellular processes and their target genes, all of which might be linked to PSP pathogenesis. However, large-scale research will have to validate this.

## Figures and Tables

**Figure 1 diagnostics-12-01204-f001:**
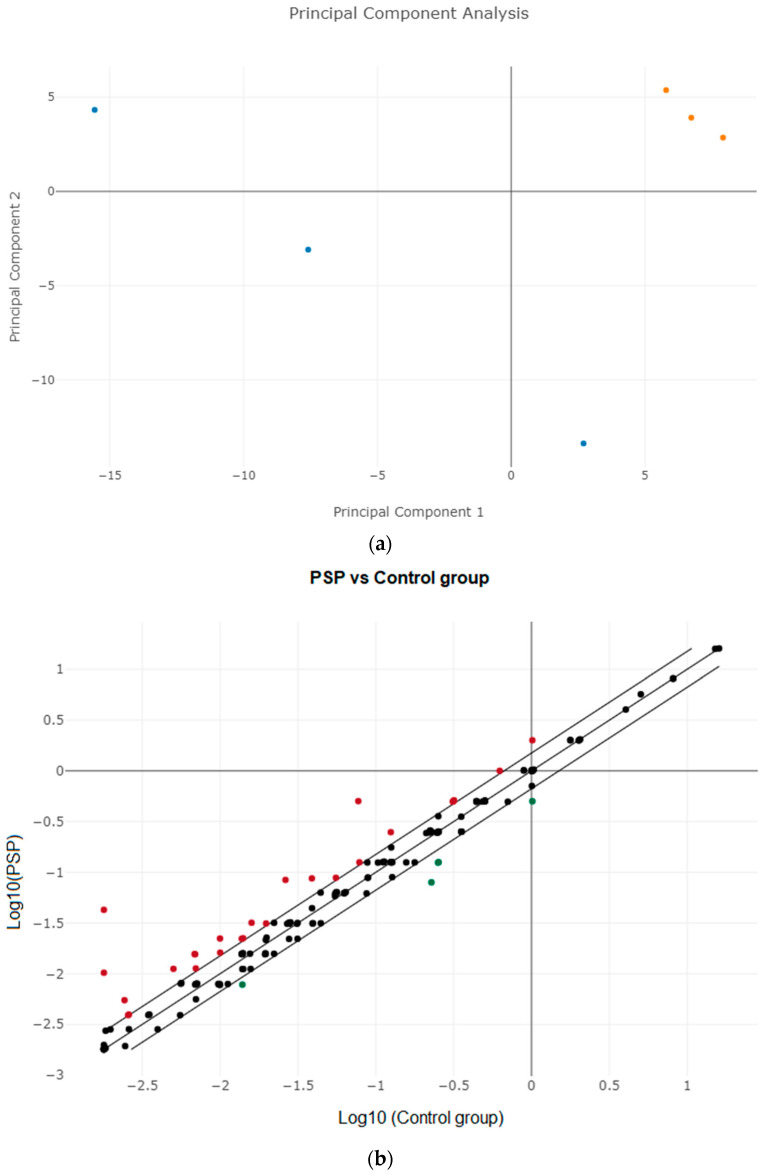
(**a**) Principal component analysis (PCA) plot. Blue-colored dots are control samples and orange-colored dots are PSP. (**b**) Scatter Plot of differentially expressed miRNAs. The up-regulated miRNAs are displayed in red, and down-regulated miRNAs are displayed in green. (**c**) Volcano plot. Red-colored dots represent overexpressed miRNAs, green-colored dots represent downregulated miRNAs compared to control, and black dots are insignificant miRNAs analyzed.

**Figure 2 diagnostics-12-01204-f002:**
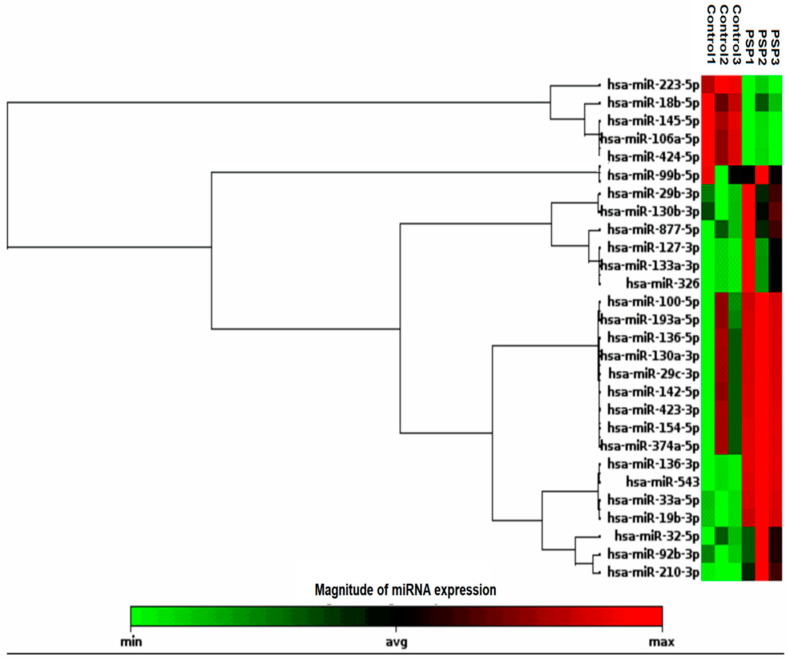
Heat map and unsupervised hierarchical clustering of plasma miRNA profiles in the study groups: The unsupervised hierarchical clustering was performed for both the control and PSP groups, and the top 28 differentially expressed miRNAs were presented in the heat map. The color scale, ranging from green to red, indicated low to high expression of each miRNA transcript.

**Table 1 diagnostics-12-01204-t001:** ROC curve analysis for miRNAs with biomarker potential and their 95% CI.

miRNAs	Profiling Study Fold Change	Validation Study Fold Change		Sensitivity (%)	Specificity (%)	AUC	95% CI	*p*-Value	Corrected *p*-Value
		Controls (n = 17) Mean ± SE	Cases (n = 18) Mean ± SE						
miR-19b-3p	2.03	1.1 ± 0.13	1.5 ± 0.11	77.78	64.71	0.7059	0.5267 to 0.8850	0.0017	0.037
miR-33a-5p	2.04	2.3 ± 0.67	6.7 ± 0.84	77.78	82.35	0.8578	0.7320 to 0.9837	0.0001	0.0003
miR-130b-3p	3.29	1.3 ± 0.33	7.3 ± 2.4	66.67	68.75	0.7778	0.6204 to 0.9352	0.0048	0.027
miR-136-3p	2.34	1.6 ± 0.39	2.9 ± 0.33	73.33	82.35	0.7882	0.6192 to 0.9573	0.0045	0.022
miR-210-3p	5.85	1.2 ± 0.19	2.2 ± 0.24	66.67	70.59	0.7810	0.6233 to 0.9387	0.0039	0.004

CI = confidence interval. ROC = Receiver operating characteristic curve; SE = standard error; AUC = area under the ROC curve.

## Data Availability

The data presented in this study are available as Appendix A.

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
