# Peer review of "Plasma microRNAs as a Potential Biomarker for Identification of Progressive Supranuclear Palsy"

_diagnostics, 2022, doi:10.3390/diagnostics12051204_

Round 1

Reviewer 1 Report

Ref: diagnostics-1714690

Title: Plasma microRNAs as a potential biomarker for identification of progressive supranuclear palsy

Manuscript Type: Original Article

Journal: Diagnostics

The authors studied 18 randomly recruited  patients (aged 60.11±1.6 years) with probable PSP within three years of the clinical diagnosis and 17 healthy individuals. From this population, we selected 12 PSP patients and 9 controls as initial screening cohort. 23 miRNAs were upregulated and five  were downregulated  in PSP patients compared to controls. In addition, expression of selected candidate miRNA was tested in 17 healthy controls and 18 PSP patients who were enrolled for the study individually. The combined ROC of these five miRNAs has an AUC of 0.7817 with a sensitivity of 72.41% and a specificity of 66.67% with a 95% of CI (0.7126 to 0.8508). Subsequent  KEGG pathway analysis revealed 48 high-risk pathways. The study revealed that as a single marker—miR-19b-3p, miR-33a-5p, miR130b-3p, miR-136-3p, and miR-210-3p had a specificity of 64.71%, 82.35%, 68.75%, 82.35% and 70.59% at sensitivity 77.78%, 77.78%, 66.67%, 73.33%, and 66.67% respectively.

The manuscript is technically sound, methods are appropriate and properly conducted, the statistical analysis of the data is sound. There are no ethical concerns.

However  some points that could be improved:

Discussion

Page 39 Five of the 23 increased miRNAs were chosen for further study: miR-210-3p, miR-19b-3p, miR-33a-5p, miR-130b-3p, and 40 miR-136-3p. There have been few results of miRNA profiles in PSP, and these miRNAs have not been documented in prior PSP investigations. Previous studies on post-mortem brain tissues discovered that miR-132 was downregulated whereas miR147a and miR-518e were upregulated

The authors should comment on why different studies (the author  should integrated the following article:   

The biomarker potential of cell-free microRNA from cerebrospinal fluid in Parkinsonian Syndromes.

Starhof C, Hejl AM, Heegaard NHH, Carlsen AL, Burton M, Lilje B, Winge K.Mov Disord. 2019 Feb;34(2):246-254. doi: 10.1002/mds.27542. Epub 2018 Dec 17; Exosomal miRNA as peripheral biomarkers in Parkinson's disease and progressive supranuclear palsy: A pilot study. Manna I, Quattrone A, De Benedittis S, Vescio B, Iaccino E, Quattrone A. Parkinsonism Relat Disord. 2021 Dec;93:77-84. A cerebrospinal fluid microRNA analysis: Progressive supranuclear palsy. Nonaka W, Takata T, Iwama H, Komatsubara S, Kobara H, Kamada M, Deguchi K, Touge T, Miyamoto O, Nakamura T, Itano T, Masaki T.Mol Med Rep. 2022 Mar;25(3):88 MicroRNA as Candidate Biomarkers in Atypical Parkinsonian Syndromes: Systematic Literature Review.

Bougea A.Medicina (Kaunas). 2022 Mar 26;58(4):483) have found different miRNAs to be characteristic of PSP.

Minor concerns

English language and typos should be carefully reviewed

Author Response

Respected editor,

Thank you for giving us the opportunity to submit a revised draft of the manuscript “Plasma microRNAs as a potential biomarker for identification of progressive supranuclear palsy” for publication in the Diagnostics. We appreciate the time and effort that you and the reviewers dedicated to providing feedback on our manuscript and are grateful for the insightful comments on and valuable improvements to our paper. We have incorporated the suggestions made by the reviewers. Those changes are highlighted within the manuscript. Please see below, for a point-by-point response to the reviewer-1 comments and concerns.

  • Discussion
  • Page 39 Five of the 23 increased miRNAs were chosen for further study: miR-210-3p, miR-19b-3p, miR-33a-5p, miR-130b-3p, and 40 miR-136-3p. There have been few results of miRNA profiles in PSP, and these miRNAs have not been documented in prior PSP investigations. Previous studies on post-mortem brain tissues discovered that miR-132 was downregulated whereas miR147a and miR-518e were upregulated
  • The authors should comment on why different studies (the author  should integrated the following article:   

The biomarker potential of cell-free microRNA from cerebrospinal fluid in Parkinsonian Syndromes. Starhof C, Hejl AM, Heegaard NHH, Carlsen AL, Burton M, Lilje B, Winge K.Mov Disord. 2019 Feb;34(2):246-254. doi: 10.1002/mds.27542. Epub 2018 Dec 17;

Exosomal miRNA as peripheral biomarkers in Parkinson's disease and progressive supranuclear palsy: A pilot study. Manna I, Quattrone A, De Benedittis S, Vescio B, Iaccino E, Quattrone A. Parkinsonism Relat Disord. 2021 Dec;93:77-84.

A cerebrospinal fluid microRNA analysis: Progressive supranuclear palsy. Nonaka W, Takata T, Iwama H, Komatsubara S, Kobara H, Kamada M, Deguchi K, Touge T, Miyamoto O, Nakamura T, Itano T, Masaki T.Mol Med Rep. 2022 Mar;25(3):88

MicroRNA as Candidate Biomarkers in Atypical Parkinsonian Syndromes: Systematic Literature Review. Bougea A.Medicina (Kaunas). 2022 Mar 26;58(4):483) have found different miRNAs to be characteristic of PSP.

Response: We thank the reviewer for the suggestion. The above articles have been cited and discussed in the revised manuscript discussion section.

  • Minor concerns

English language and typos should be carefully reviewed

Response: We thank the reviewer for the suggestion, we have taken input from the English language expert and typos have been taken care of.

Reviewer 2 Report

The manuscript titled with “Plasma microRNAs as a potential biomarker for identification of progressive supranuclear palsy” itself is interesting and is suitable for this journal. In this study, however, there are few key remarks when it comes to the coverage and explanation.

Comments to the Author

The manuscript titled with “Plasma microRNAs as a potential biomarker for identification of progressive supranuclear palsy” itself is interesting and is suitable for this journal. In this study, however, there are few key remarks when it comes to the coverage and explanation.

The authors did not clearly explain in the given below paragraph:

  1. Authors should elaborate the sample collection procedure or add reference.
  2. Authors not contributed about the TLC of the patient sample, before to start plasma separation. Is it important or not for this study?
  3. How authors quantify or check the quality of the miRNA, explain?
  4. In section 2.3, few lines should be the part of results.
  5. In section 2.5, add few more references or explain more.
  6. In the result part qPCR result analysis is not explained separately, but it is mentioned in the material and methods section, why?
  7. Image quality of figure 1 is poor.
  8. Authors mention “the exact diagnosis and subtyping of PSP, a more reliable, non-invasive, cost-effective, and disease-relevant biomarker is required”. Can authors performed any experiment or compare his study with any other study to justify the above said line?

Author Response

Respected editor,

Thank you for giving us the opportunity to submit a revised draft of the manuscript “Plasma microRNAs as a potential biomarker for identification of progressive supranuclear palsy” for publication in the Diagnostics. We appreciate the time and effort that you and the reviewers dedicated to providing feedback on our manuscript and are grateful for the insightful comments on and valuable improvements to our paper. We have incorporated the suggestions made by the reviewers. Those changes are highlighted within the manuscript. Please see below, for a point-by-point response to the reviewer-2 comments and concerns.

  1. Authors should elaborate the sample collection procedure or add reference.

Response: We thank the reviewer for the suggestion. We have elaborated the sample collection procedure in the revised manuscript.

  1. Authors not contributed about the TLC of the patient sample, before to start plasma separation. Is it important or not for this study?

Response: TLC of the patient sample is not done and it is not required for this study.

  1. How authors quantify or check the quality of the miRNA, explain?

Response: We thank the reviewer for the suggestion. We have updated the procedure to check the quality of miRNA in the revised manuscript.

  1. In section 2.3, few lines should be the part of results.

Response: Appropriate changes have been made in the revised manuscript as suggested.

  1. In section 2.5, add few more references or explain more.

Response: We thank the reviewer for the suggestion. We have incorporated the reference in the revised manuscript as suggested.

  1. In the result part qPCR result analysis is not explained separately, but it is mentioned in the material and methods section, why?

Response: The result part of the qPCR analysis section continued with the profiling result earlier, now it has been separated with the heading “3.3 Validation of candidate miRNAs using RT-qPCR” in the revised manuscript.

  1. Image quality of figure 1 is poor.

Response: We thank the reviewer for the suggestion. We have incorporated good quality (600DPI) Figure 1 as suggested.

  1. Authors mention “the exact diagnosis and subtyping of PSP, a more reliable, non-invasive, cost-effective, and disease-relevant biomarker is required”. Can authors performed any experiment or compare his study with any other study to justify the above said line?

Response: We have quantified serum NFL as a biomarker to identify and differentiate PD from PSP-P and MSA-P. (Yadav R, Ramaswamy P, Pal P, Christopher R. Serum neurofilament light chain differentiates early Parkinson's disease from progressive supranuclear palsy-parkinsonism. In MOVEMENT DISORDERS 2020 Sep 1 (Vol. 35, pp. S520-S521). 111 RIVER ST, HOBOKEN 07030-5774, NJ USA: WILEY).Differentiating PD from PSP-P could be accurately possible with serum analysis of NFL, but not from the MSA-P. As well as in our study we were unable to differentiate PSP-P from healthy controls. This could be due to small sample size, however could not be ignored. As well as other studies have been discussed in the revised manuscript discussion section.
